# Are carbon emissions trading and green financial instruments synergistic? -Comprehensive quantitative research based on content analysis

**Hongjian Yu[1], Xiufan Zhang[2]\***

1 Alibaba Business School, Hangzhou Normal University, Hangzhou, China, 2 School of Economics and Management, Zhejiang Sci-Tech University, Hangzhou, China

\* zhangxiufan@zstu.edu.cn

**Data Availability Statement:** All relevant data are within the manuscript and its Supporting information files.

## Abstract

Coordinating policies is an essential guarantee for carbon emission reduction and sustainable development. Based on the theoretical framework of the policy paradigm, we quantitatively analyze 266 policy documents on promoting carbon emission trading and green financial policies from 2011 to 2022 using the content analysis research method. Based on the matching network of "policy objectives-policy tools," we analyze the synergistic characteristics of carbon emission trading policies and green financial policies in promoting carbon emission reduction targets and reveal the matching mode of "objectives-tools" of green financial policies by using social network analysis. It is found that, first, from the perspective of policy objectives, the main policy objectives of carbon emissions trading are to promote green innovation of enterprises, and the main policy objectives of green finance are to promote green development, which reflects the consistency and endogenous motivation of policy objectives. Secondly, command-control and market incentive policy tools are the main policy tools in the structure of policy tools. The proportion of public participation policy tools is small, and there is a structural asymmetry. Third, carbon emissions trading tools focus on supervision, adjustment, and platform construction. The green financial policy tools have the characteristics of guidance, public welfare, and externality. The two constitute a complementary, embedded, and integrated ' double synergy ' carbon emission reduction policy. Based on this, this paper puts forward some suggestions to promote policy coordination and provides a reference for China to achieve the dual carbon goal.

## Introduction

It is of great significance to establish an efficient linkage mechanism of policy coordination for improving enterprises' green innovation capability and the overall carbon emission reduction level [1]. At present, China's carbon emission trading mechanism and green financial policy play an important role in achieving the dual carbon goals of carbon peaking and carbon

**Funding:** This research was funded by the following foundations: National Natural Science Foundation (grant number: 72304250); Philosophical Social Planning of Zhejiang Province (grant number: 24NDQN093YB). The funders had no role in study design, data collection and analysis, decision to publish, or preparation of the manuscript.

**Competing interests:** The authors have declared that no competing interests exist.

neutralization, which has become a market consensus. Among them, carbon finance is an integral part of green finance, which refers to various financial institutional arrangements and financial trading activities aimed at reducing greenhouse gas emissions, mainly including trading and investment of carbon emission rights and their derivatives, investment and financing of low-carbon project development and other related financial intermediary activities. There may be a synergy between carbon emission rights trading and green financial policy objectives and tools [2]. Currently, China's carbon emission reduction process still needs some problems, such as insufficient motivation of energy users to reduce emissions, imperfect support system of green high-tech and intelligent technology, and insufficient awareness of collaborative emission reduction. The corresponding policy system must still be continuously innovated and improved [3, 4]. Carbon emission reduction and urban green governance are complicated systematic projects requiring high system coordination. At the same time, the formulation and implementation of policies often have particular uncertainty and subjectivity, which shape the endogenous power of policy coordination. The internal relationship between carbon emission trading and China's green finance has attracted much attention from academic circles. A large number of studies have shown that "green finance" takes environmental protection and effective utilization of resources as one of the criteria to measure the effectiveness of its activities and guides the coordinated development of financial activities, environmental protection, and ecological balance [5]. The carbon emission trading mechanism includes critical emission units, guides market transactions by setting emission quotas, and achieves overall emission reduction targets through market performance and supervision [6] "Green finance" mainly includes green credit, green securities, and green insurance. Green finance can guide funds to gradually withdraw from industries with high energy consumption and high pollution and invest more in green and environmental protection industries, a powerful guarantee for promoting green innovation and industrial structure transformation. However, as a new financial format, green finance faces problems such as complex internalization of positive externalities and asymmetric green information, which hinder its development. The national carbon emission trading market is under construction. Compared with other mature developed countries, there is still a significant market maturity, liquidity, and trading scale gap. Therefore, it is essential to promote green finance development through government departments' green financial policies [7]. China's green finance is still based on credit, and green insurance and green securities are in the exploration and initial stages. In green finance development, carbon finance and other tools promote the construction of a "carbon emission trading market."

It is worth noting that promoting urban green development must be integrated with the coordination of carbon emission trading policy and green financial policy. Policy Coordination is essential to give full play to the actual effect of a complex Policy Mix. The existing literature focuses on the vertical transmission of policy-making in a specific field under the vertical management system of the government and the horizontal coordination of policies in regions or departments [8]. Most of the studies on policy synergy are based on inter-governmental relations, departmental or regional cooperation with single-field policy combinations, and policy transmission under a multi-level administrative system [9]. However, few pieces of literature pay attention to the synergistic characteristics of policy mix, such as tool-goal-subject. Currently, most of the related literature on carbon emission reduction policy puts forward relevant policy suggestions based on theoretical and empirical analysis. In contrast, only some existing policies are decomposed in refined grains. The Kyoto Protocol is stimulating the development of emissions trading schemes at the national and international levels. The release of multi-agent policies, such as green credit policies, green insurance policies, carbon emission trading policies, and carbon finance policies, has led to complex policy interaction problems [10]. The need for more evidence and assessment practice on policy mix and interaction

reflects the challenges of the interaction of analytical tools and the widespread neglect of the consideration of interactions in policy formulation. However, policy interaction or synergy has yet to be thoroughly studied. The research on carbon emission trading, green finance, and their synergistic effects on carbon emission reduction has rich theoretical value and practical significance. Direct research on the synergistic effects of carbon emission trading and green finance on carbon emission reduction is rare, and it is urgent to follow up the research and exploration with Chinese local characteristics. We put the concept of policy synergy on the "policy tool-goal" synergy, take the synergy between carbon emission trading and green financial policy as the research object, and investigate the synergy characteristics between carbon emission trading policy and green financial policy, which has reference significance for perfecting China's carbon emission reduction policy. In October 2011, the General Office of the National Development and Reform Commission issued the Notice on Piloting Carbon Emission Trading. On February 24, 2012, the China Banking Regulatory Commission issued the Green Credit Guidelines (Y.B.F. [2012] No.4, from now on referred to as the Guidelines. This paper selects the release of the two landmark central policy documents as the starting point for promoting carbon emission trading and green finance, respectively. The collection of policy texts ends in 2022 to investigate policy synergy's characteristics and action path. The policy documents come from the carbon emission trading network, the Chinese government network's policy document database, and various ministries and commissions under the State Council. We select the policy texts of the C.P.C. Central Committee, the State Council, and various departments on promoting carbon emission trading and green finance from 2011 to 2022 as the research objects. After the fine-grained decomposition of the policy texts, a matching network of "policy objectives-policy tools" is constructed to investigate the synergistic characteristics of the policy combination of carbon emission trading and green finance.

To sum up, our possible innovations and contributions are reflected in three aspects. First, further enrich the research on policy mix from the perspective of policy synergy. We apply the concept of policy synergy to analyze the interactive characteristics of carbon emissions trading and green financial policies, which is helpful to understand the policy logic when the government matches goals and tools in the process of carbon emission reduction and the inter-governmental synergy among multi-agents. Secondly, by systematically combing the related research on carbon emission trading and green finance and achieving carbon emission reduction targets, we use qualitative and quantitative methods to analyze the coordination of internal and external policies deeply. Based on the matching network of the "policy objective-policy tool," we adopt the methods of quantitative analysis of differences, visual analysis of policy network, and qualitative analysis of policy text. It proves the synergy between carbon emission trading and green finance from a policy perspective. Thirdly, for the classification of policy tools, the mainstream way in academic circles is to adopt the classification standard of Rothwell & Zegveld (1984) and divide policy tools into supply, demand, and environment types [11]. Based on this, literature has conducted extended research in innovation and sustainable development. According to the characteristics of green financial policy and carbon emission reduction policy, we divide policy tools into "command control," "market incentive," and "public participation," builds a carbon emission reduction policy based on the existing policy network, and provides more clear policy objectives for policymakers.

The arrangement of this paper is as follows: Based on the above analysis, we have the following three research objectives. The first research goal is to summarize the core objectives and tools of green financial policy and carbon emission trading policy by counting the word frequency of 266 policy documents according to the policy system. The second research goal is to investigate the synergy characteristics of carbon emissions trading and green financial policy systems through policy synergy analysis, using three methods:

quantitative analysis of differences, visualization of a policy network, and qualitative analysis of policy text in the merged policy network. The third research goal is to research conclusions and policy recommendations and analyze the direction of using green finance and carbon emission trading policies to guide collaborative emission reduction exhibitions in the future.

## Literature review

### Carbon trading policy

Under the market mechanism, market entities with excess or insufficient quotas use the carbon emission trading market to trade, which can effectively reduce the overall cost of carbon emission reduction while achieving the overall goal of carbon emission reduction. By setting the total carbon emission control policy, the government will incorporate the emission cost into the transaction cost of enterprises to achieve environmental optimization [12]. Among the 27 Member States of the European Union, the urban clusters of some member States have further improved the system and regulations of carbon emission trading mechanism, promoted low-carbon technology innovation, and actively exerted the emission reduction effect [13]. Zhang et al. (2020) explore the emission reduction effect, economic effect, and carbon market efficiency of China's carbon emission trading policy, thus deeply exploring the policy effect of the carbon emission trading system and further analyzing the efficiency impact of the carbon emission trading system on economy and environment [14]. China's carbon emission trading (CET) policy aims to force related enterprises to implement low-carbon technological innovation in constructing carbon emission trading systems through market-oriented means [15]. China's carbon emission trading (CET) policy helps to improve the technological innovation capability of power and aviation enterprises. The quota allocation and performance links strengthen the carbon emission restrictions on high energy-consuming products of business owners, and the voluntary emission reduction offset mechanism encourages enterprises to voluntarily implement emission reduction projects [16]. The National Development and Reform Commission supervises the voluntary emission reduction offset mechanism, and emission control enterprises can obtain specific national certified voluntary emission reduction (CCER) after being certified by relevant authorities. CCER refers to the carbon emission reduction offset formed by the Nanning Project, a fully automatic dosing device such as renewable energy and forestry carbon sink in China [17]. The voluntary emission reduction offset mechanism complements the carbon trading market, including offset ratio, identification process, project feasibility assessment, and emission reduction calculation [18, 19].

Liu et al. (2022) take China's carbon emission trading policy (CETP) as the research object to explore the impact of CETP on the synergistic benefits of carbon emission reduction and air pollution control [1]. Based on the city level, we discuss carbon emission trading policy, carbon finance, and carbon emission reduction. It is found that the implementation of a carbon emissions trading policy (CETP) promotes carbon finance, thus reducing carbon emissions. This effect has spillover. Thus, the essence of carbon emission trading policy is an environmental and economic policy aiming to control the total amount of carbon emissions. The government encourages enterprises to take the initiative to reduce carbon emissions and actively participate in market transactions.

### Green financial policy

With the increasing global concern for environmental protection, climate change, and sustainable development, policymakers and researchers have recently focused on green finance. The green financial reform and innovation pilot zone builds an innovative green financial industry,

develops a project library, and gradually improves a unified management platform [20]. Under the background of carbon peak and carbon neutrality, it is more and more important to effectively improve green competitiveness and realize digital green innovation and development [21]. The government's green financial policy is inclined to environmental protection enterprises. The low-carbon technology capability of enterprises in manufacturing links is increasing daily, and the energy consumption and carbon emission intensity are gradually decreasing. Enterprises further develop emission reduction capabilities [22]. Green credit policy can restrain the credit demand of polluting enterprises such as high carbon, eliminate backward production capacity, impose emission reduction constraints on high-emission enterprises, and improve their sense of social responsibility by guiding the financing needs of environmental protection projects such as energy conservation [23]. However, some studies believe that green financial policies combine financial and environmental policies, and the emission reduction constraints imposed by environmental policies on high-emission enterprises may inhibit innovation and development, which is not conducive to the long-term development process of enterprises [24–26]. For example, Zhang et al. (2022) explore how to control air pollution by economic means and explore China's green financial policy [23, 27]. Their results show that green financial policies have the effect of reducing carbon emissions. According to different pollution sources, the effects of green financial policies are different. Akomea et al. (2022) analyze and summarize the green financial policies through content analysis and finds that green securities, green investment, climate finance, carbon finance, green insurance, green credit, and green infrastructure bonds are part of banks' critical green financial products [28]. It is of great significance to evaluate whether green credit policy can promote green development. China's green credit policy has environmental effects, thus boosting green development. Green finance policy plays a role in carbon neutral deployment of renewable energy and puts forward green financing policy in emission reduction deployment of renewable energy [29]. The green financial policy affects and increases the debt financing cost of heavily polluting enterprises. Therefore, the green financial system should improve further [30]. Therefore, the supporting policies, measures, and development strategies of green finance are major mechanism innovations and policy tools to promote the transformation and development of the low-carbon economy in China.

## Policy mix

The policy mix is to take optimization measures such as coordination, trade-off, and compromise on the choice of relevant policy tools [31]. According to the complexity of policy formulation and implementation, combining different policy tools forms the optimal policy combination theory [32]. Many researchers and policymakers have called for the best policy mix to address significant issues such as climate change and biodiversity conservation. Policy formulation is often adjusted with the changes in social, economic, political, and organizational background, reflecting the "systematic" characteristics of the policy mix. Some scholars put forward the definition of policy combination tension and interaction from the perspective of national policy combination. It is considered that the practical exertion of policy effect depends on the close cooperation of different types of policy tools, which has enlightenment significance for future prescriptive and analytical policy research [33, 34]. Lehmann (2012) provides policy combinations for pollution control, analyzing the use of multiple policy combinations to address a single pollution problem [35]. The combination of energy efficiency and policy proposed by Rosenow et al. (2016) enriches the theoretical literature on policy combination. It empirically analyzes the policy combination of building efficiency policies in 14 EU countries [36]. Through a combination of policy tools, energy efficiency policies are expected

to play a vital role in achieving EU energy goals, especially in reducing energy demand and $CO_2$ emissions. Transaction costs are integrated into analyzing pollution problems and policies to overcome pollution problems. The policy mix may help correct multiple strengthening failures of private governance structures, such as pollution externalities and technology spillovers.

When defining policy combinations, we should distinguish between policy objectives and tool combinations. The rationale for combining policy objectives exists in distribution and other political issues. In contrast, the rationale for combining policy instruments is mainly related to specific markets, governance, or behavioral failures. Therefore, promoting sustainable development requires cooperating with different policy tools [37, 38]. For example, carbon emission trading mechanisms can interact with carbon/energy tax, renewable energy power support mechanisms, and policies to promote energy efficiency. At the level of inter-governmental synergy, the policy mix has more complex interactions with the increase of administrative levels. Interaction effects based on intergovernmental relations include Top-down and Bottom-up policy diffusion, regional or inter-sectoral competition and cooperation, such as the design combination of cross-policy institutions or the combination of specific institutions, and the experience of policy combination practice. Intergovernmental Relations (IGR), also known as "inter-governmental relations," includes not only the vertical relations between central government departments and local governments but also the division of power among departments within local governments at the same level and the horizontal relations between local governments at different levels and government departments in different regions [39]. Intergovernmental cooperation is a cooperative relationship between governments at all levels to formulate and implement policies or provide services [40, 41]. The cooperation between governments at all levels is more reflected as a kind of mutual synergy, which can better optimize the efficiency of regional resource allocation and achieve the goal of multi-subject win-win [42]. The complementarity, mutual trust, and resource superposition of green innovation technologies make digital green innovation knowledge transfer from partners to enterprises [43]. By adjusting and integrating the powers and interests among policy subjects, we can form consistent goals and actions, reduce policy conflicts, frictions, and failures caused by externalities and transaction costs, improve policy operation efficiency, and promote the solution of complex cross-border problems.

By combing the existing literature, according to the coordination characteristics of the policy itself, to optimize the allocation of resource elements, the existing literature uses the event analysis method to identify the cause and effect of policy synergy. The synergy indicators are constructed based on the quantitative characteristics of policies in different dimensions, such as policy principles, implementation modes, policy actors, and policy tools. Alternatively, decompose the policy text in a fine-grained way, investigate the coordination relationship between multiple policy subjects and different policy tools in terms of objectives and behaviors under the framework of the policy network, analyze the characteristics of inter-governmental horizontal cooperation at different stages, and find the evolution trend of China's inter-governmental cooperation network towards complexity and centralization. From the perspective of causal identification, the correlation between carbon emissions trading and China's green finance is empirically tested, and the synergy between them is rarely described and analyzed based on the content characteristics of policy texts. Therefore, we apply policy synergy to analyze the interactive characteristics of carbon emission trading and green finance, uses qualitative and quantitative methods to analyze the relevant policies to promote carbon emission reduction deeply, and investigates the synergy of the policy system in three aspects: objectives, tools, and subjects.

## Experimental

### Research methods

Different types of policies have different mechanisms. With the establishment and improvement of digital government affairs and policy document databases of governments at all levels in China, the text analysis method has been widely used in public policy research and related fields [44]. Digital technology has realized the whole process of green manufacturing and broken the space barrier [45]. With the help of the title, subject, time, subject words, document reference relationship, and other characteristics of policy texts, we compare the attitudes, positions, and behavior laws of different subjects in time and space. From the perspective of policy tools, we investigate the characteristics of policy tools and their combinations, such as the distribution characteristics and international comparison of policy tools, the matching between policy tools and policy objectives, and policy synergy [46, 47]. Policy objectives and tools are two dimensions often used in policy analysis [48, 49]. The construction of a "policy goal-policy tool" matching network can be divided into three steps.

Text recognition. Identify policy documents' introductory paragraphs and "keywords" paragraphs and create a "verb/noun" thesaurus related to policy objectives/tools. Highly similar policy objectives/policy tools are uniformly expressed according to the sentence structure of "verb + noun."

### Integrated frame of policy analysis

Based on the constructed 'policy objectives-policy tools' dual policy network structure, we analyze the carbon emission trading and green financial policy texts from four dimensions, as shown in Fig 1.

The regional spatial hierarchy dimension divides the policy text into provincial spatial levels according to the administrative level of the policy issuing unit. The policy release time covers 2011–2022, reflecting the changes in the number of policies and themes over time. The policy content dimension is based on the specific policy content in the text.

### Data collection and processing

We collect the policy documents related to green finance issued by the State Council, ministries, provinces, autonomous regions, and municipalities directly under the Central Government from 2011 to 2022. The types of policy texts include opinions, notices, guidelines, announcements, outlines, plans, programs, and government work reports. To ensure the comprehensiveness and accuracy of the data, we use keywords for full-text search in the collection process, such as "carbon emission trading," "carbon trading," "green finance," "carbon finance," "green credit," and "green bond," and documents such as "letter," "Mingdian," "request for instructions" and "reply" are filtered. To improve the correlation between policy texts and policy themes, after reading the policy contents preliminarily, the policy documents with low correlation were removed, and finally, 266 policy texts were obtained, including 110 texts of the carbon emission trading policy and 156 texts of the green financial policy system.

## Results and analysis

### Characteristics of policy text

According to the 266 policy texts collected in this paper, the excerpts of policy texts are sorted out as shown in Table 1. The issue of carbon quotas involves extremely complex factors, involving enterprises, markets, competent authorities, and other subjects, as well as the results

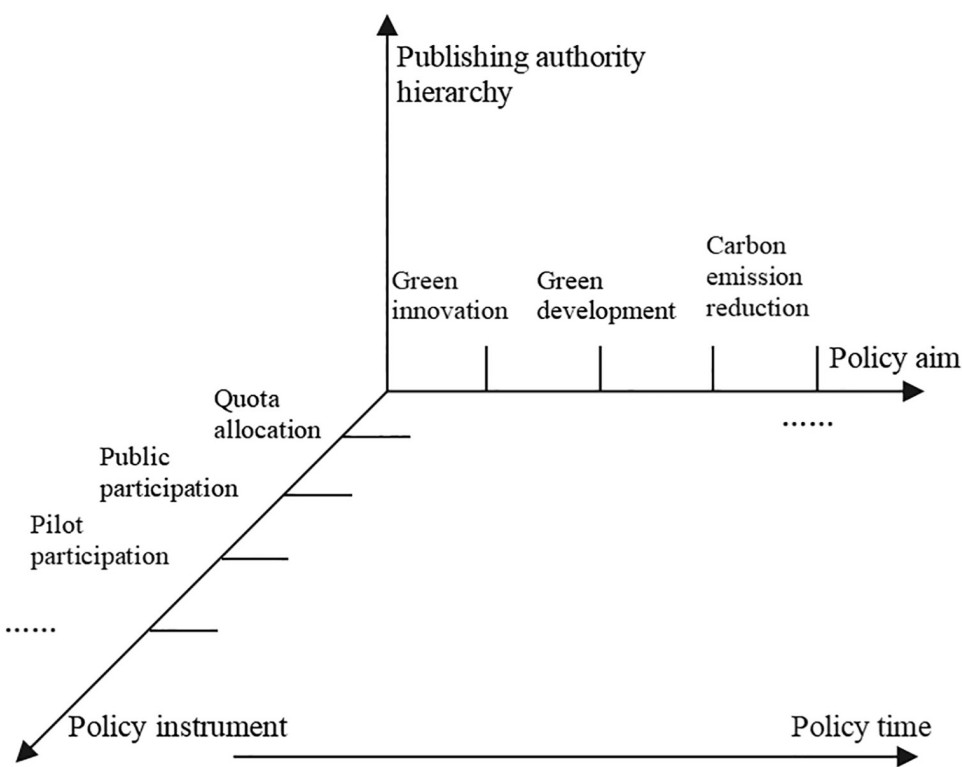

**Fig 1. Carbon emissions trading and green financial policy collaborative research framework.** Source: Self-painted by the author.

**Table 1. Excerpts of policy text.**

| Number | Level | Year | File name | Policy priorities |
|---|---|---|---|---|
| 1 | National Development and Reform Commission | 2011 | 'National Development and Reform Commission on the pilot work of carbon emissions trading notice' | Carbon trading |
| 2 | State Council | 2011 | Notice of the '12th Five-Year' Energy Saving and Emission Reduction Comprehensive Work Plan' | Carbon trading |
| 3 | China Banking and Insurance Regulatory Commission | 2011 | 'Bank of Communications 2011 industry credit investment guidelines' | Green finance |
| ... | ... | ... | ... | ... |
| 71 | Hubei Province | 2014 | 'Hubei Province carbon emissions trading pilot work implementation plan' | Carbon trading |
| 72 | Guangdong Province | 2014 | " Detailed Rules for the Implementation of Carbon Emission Quota Management in Guangdong Province (Trial) " | Carbon trading |
| ... | ... | ... | ... | ... |
| 126 | Central Bank and other ministries | 2016 | 'Guidance on building a green financial system' | Green finance |
| 127 | National Development and Reform Commission | 2016 | Notice on Doing a Good Job in Launching the National Carbon Emissions Trading Market | Carbon trading |
| ... | ... | ... | ... | ... |
| 265 | People's Bank of China | 2021 | 'Green financial evaluation scheme of banking financial institutions' | Green finance |
| 266 | National Development and Reform Commission | 2021 | National carbon emissions trading management approach (Trial) | Carbon trading |

Source: Organized by the author

of carbon emissions, carbon trading, control, quota management, total amount, issuance, and stage division. In terms of the main body of policy formulation, the main body of policy issuance presents diversified characteristics, mainly the National Development and Reform Commission, the Ministry of Environmental Protection, and the People's Bank, supplemented by other departments, all of which have the enthusiasm to develop green finance and carbon trading. Especially since 2016, China's responsibility as a major country in the world has increased significantly. It has deepened the concept of green development, actively joined the global environmental governance, established a green financial system, and deepened the carbon emission trading mechanism, making a top-level design for the development of green finance and carbon trading. Driven by national policies, pilot provinces and other regions in China have also issued local development requirements such as green finance and carbon emission reduction, and put forward further plans.

By sorting out the policy text, it can be seen that China's seven carbon emission trading markets have pointed out the importance of carbon trading pilots by formulating relevant policies, setting the overall arrangement of carbon trading pilot construction, key tasks (quota management, carbon emission verification, and quota clearance). The basic process (quota trading, supervision, and guarantee), safeguard measures, and responsibilities of each participant form policy guidance and support measures. The green financial policy encourages financial institutions to incorporate environmental, social, and governance requirements into business processes, improve the level of green financial professional services for private energy-saving and environmental protection enterprises, and vigorously develop green financing. Encourage financial institutions to incorporate environmental, social, and governance requirements into business processes, improve the professional service level of green finance for private energy-saving and environmental protection enterprises, and vigorously develop green financing. Technological transformation is the most direct and effective means to promote the transformation and upgrading of traditional industries. Green finance supports the policy of real enterprises, deepens the cooperation between enterprises and banks, and strives for the optimal financial support service policy.

## Time dimension analysis

From 2011 to 2021, the total number of relevant policies generally showed a continuous upward trend, and the comparison chart of the number of central and local policy releases is shown in Fig 2. From the overall level of coordination between carbon emissions trading and green finance, the "Notice of the National Development and Reform Commission on the Pilot Work of Carbon Emissions Trading" gives a top-level design plan for carbon trading. In 2012, the China Banking Regulatory Commission issued No.4 to formulate "Green Credit Guidelines" to guide the development of green finance. As a result, an overall guiding policy for carbon emission trading and green finance to promote green development has been gradually formed. From the provincial level, the annual number of policy texts in the carbon trading pilot market is large, and the difference in changes is small. Continue to pay attention to the construction and improvement of the carbon emission trading market. A comparison of the number of central and local policy releases is shown in Fig 2.

From the perspective of the main body, the number of local policy releases is higher than that of the central policy. In view of the continuous construction and improvement of the green financial reform and innovation pilot area and the carbon emission trading pilot market, local governments need to constantly formulate relevant policies to guide the development of the market. Different regions can also use the opportunity of coordinated development of carbon emission trading and green finance to improve their own green development level.

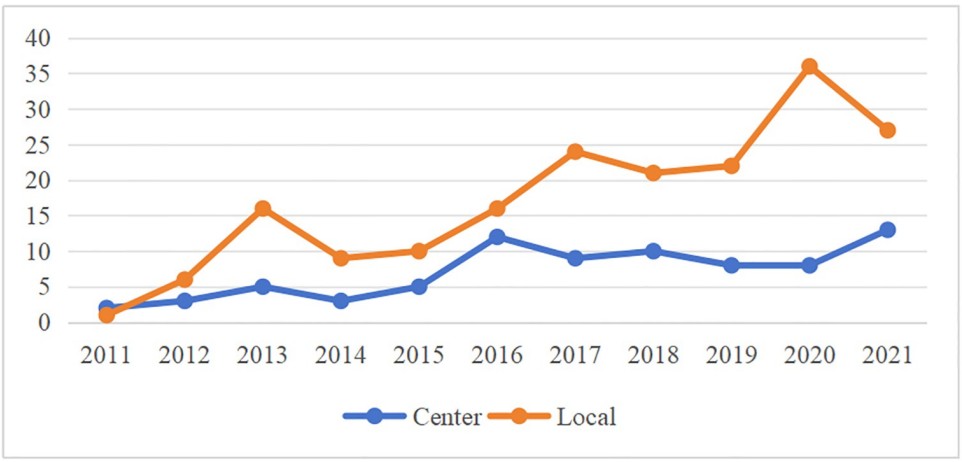

**Fig 2. Comparison of the number of central and local policy releases.**

## Policy focus

The content features of policy texts are mainly expressed by their complex semantic features. Word frequency features can help understand the policy content and high-frequency words can represent the policy focus. 266 policy documents are counted according to the policy system, and word clouds are drawn, as shown in Figs 3 and 4.

The high-frequency vocabulary is divided into policy objects and policy behaviors, and the word frequency statistics of promoting carbon emission trading policies are carried out

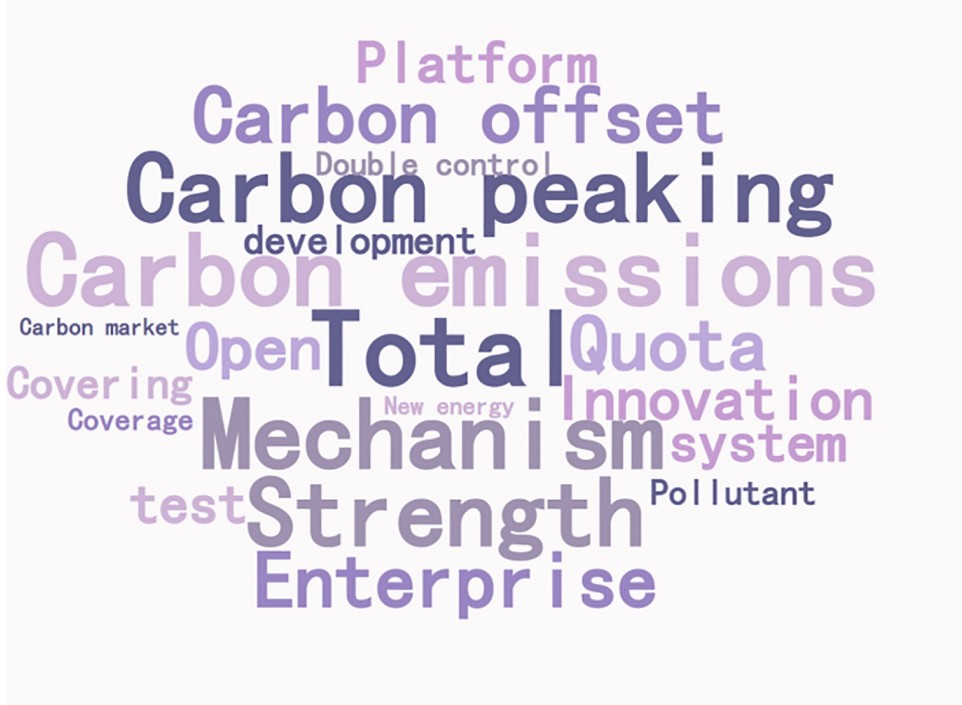

**Fig 3. Text word cloud of promoting carbon emission trading policy.**

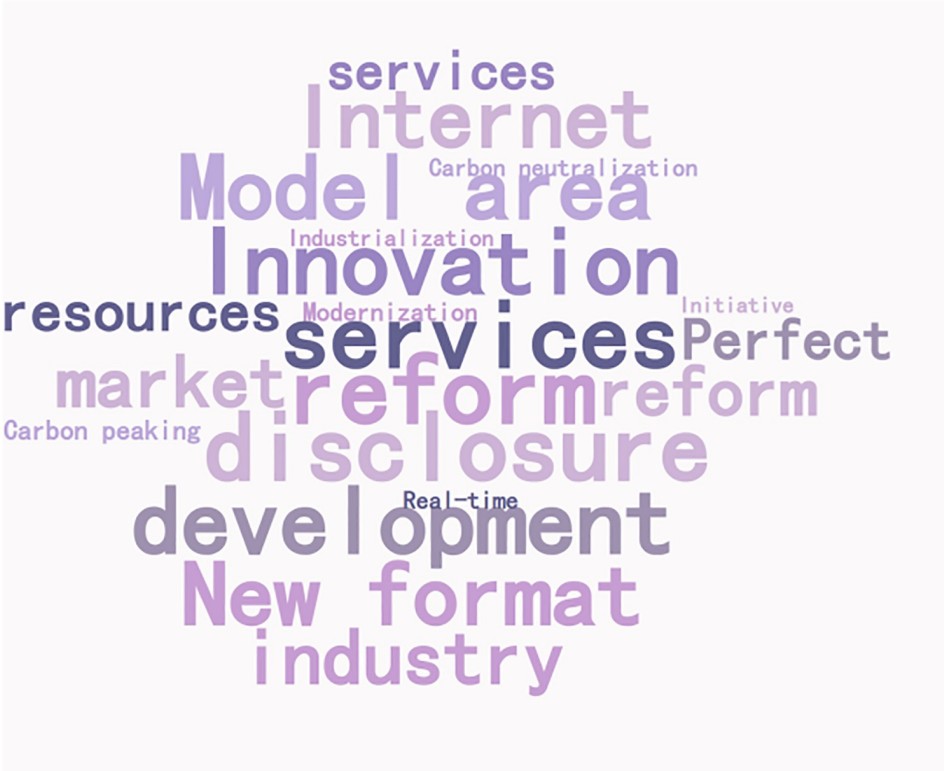

**Fig 4. Promoting the text word cloud of green financial policy.**

according to two words and three words respectively, as shown in Table 2. The core words and network connections of carbon trading policy content are shown in Fig 5.

In the vocabulary of policy objects of carbon emission trading, the top three words are "mechanism," "pilot," and "total amount," and the top three words are "pollutant," "coverage," and "service industry"; In the vocabulary of policy behavior, the top three words are "double control," "development" and "coverage," and the top three words are "carbon trading," "carbon emission" and "carbon peak". Divide high-frequency vocabulary into policy objects and behaviors, and promote green financial policies to make word frequency statistics according to two-word and three-word, respectively, as shown in Table 3. The core words and network links of green financial policy content are shown in Fig 6.

In the vocabulary of policy objects of green finance, the top three words are "technology," "enterprise," and "government," and the top three words are "financial industry," "new energy," and "experimental area"; In the vocabulary of policy behavior, the top three words are "innovation," "development" and "service"; The top three words are "digitalization," "informationization" and "diversification."

## Core policy objectives and policy tools

We identify 18 policy objectives and 35 policy tools to promote carbon emission trading, forming 268 matching combinations of "policy objectives and policy tools." There are 25 policy objectives and 56 policy tools to promote green finance, forming 858 matching combinations of "policy objectives and policy tools."

**Table 2. Promoting carbon emission trading policies (Chinese words) (Top 10).**

| | Target of carbon emission trading policy | | | | | | Policy behavior of carbon emission trading | | | | | |
| Two-character word | Frequency/ times | Proportion/ % | Three-character word | Frequency/ times | Proportion/ % | Two-character word | Frequency/ times | Proportion/ % | Three-character word | Frequency/ times | Proportion/ % |
|---|---|---|---|---|---|---|---|---|---|---|---|
| Pilot | 1800 | 1.86 | Pollutant | 281 | 0.29 | Open | 897 | 0.97 | Carbon emissions | 1617 | 1.75 |
| Total | 1693 | 1.75 | Coverage | 193 | 0.20 | Innovation | 826 | 0.89 | Carbon peaking | 1524 | 1.65 |
| Mechanism | 1549 | 1.13 | Service industry | 170 | 0.18 | management | 810 | 0.88 | Carbon emission reduction | 1515 | 1.64 |
| Strength | 1503 | 1.55 | Carbon account | 167 | 0.17 | test | 700 | 0.76 | Carbon offset | 1153 | 1.25 |
| Enterprise | 1126 | 1.60 | Demonstration area | 166 | 0.17 | supervision | 643 | 0.70 | Carbon trading | 1119 | 1.21 |
| Quota | 1093 | 1.16 | Carbon footprint | 166 | 0.17 | Certification | 572 | 0.62 | Digitization | 999 | 1.08 |
| Platform | 742 | 0.77 | Carbon finance | 161 | 0.17 | construction | 494 | 0.54 | Intelligent | 880 | 0.95 |
| Carbon sink | 735 | 0.74 | The whole society | 139 | 0.14 | Covering | 477 | 0.52 | Marketization | 801 | 0.87 |
| Carbon tax | 731 | 0.76 | New energy | 92 | 0.09 | development | 395 | 0.43 | Greening | 783 | 0.85 |
| system | 715 | 0.75 | Carbon market | 70 | 0.07 | Double control | 232 | 0.25 | Low carbonization | 778 | 0.84 |

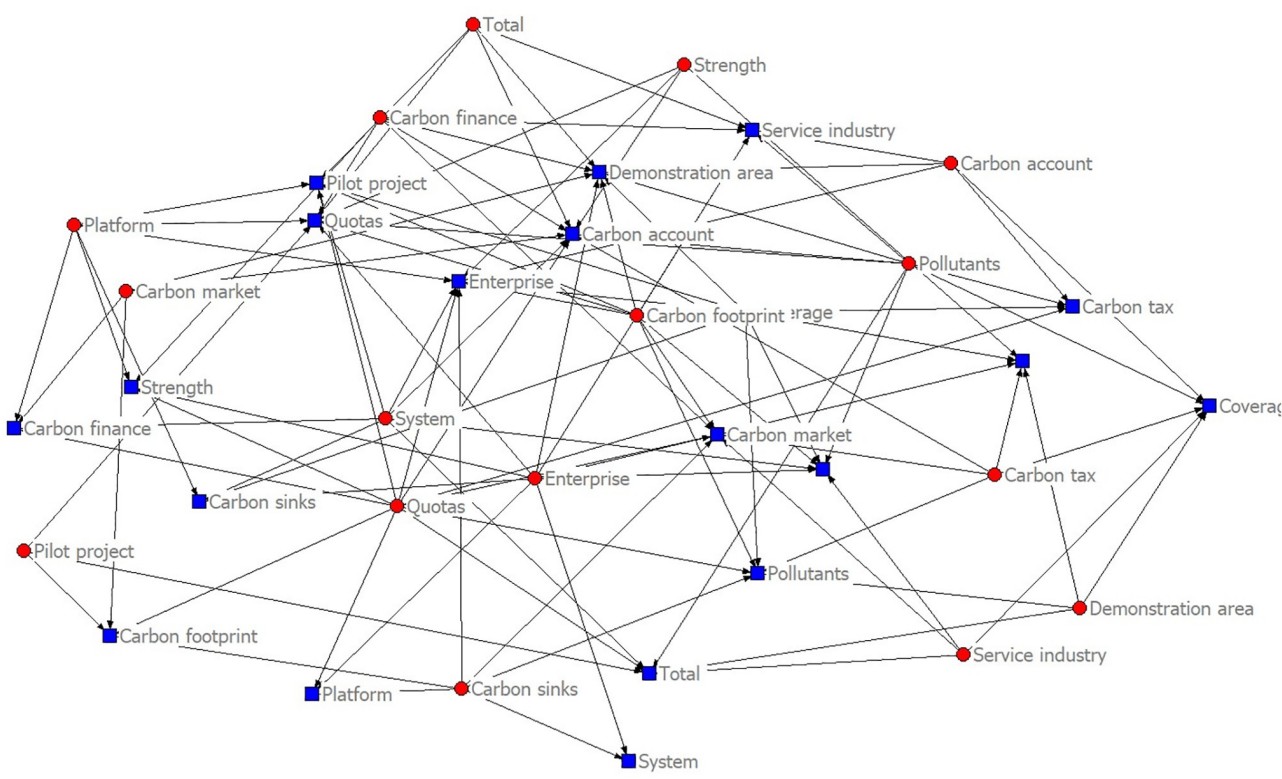

**Fig 5. Carbon trading policy content core words and network connection diagram.**

**1. Core policy objectives.** Among the policy objectives of carbon emission trading, the highest score of feature vector centrality is "promoting green innovation of enterprises," such as the Interim Measures for the Administration of Carbon Emission Trading promulgated by China's carbon emission trading pilot market and the Carbon Emission Quota Allocation Plan (Draft for Comment) drafted by the Ecological Environment Bureau (referred to as the "Plan") all point out that carbon emission trading focuses on steel, chemical, petrochemical and other industries. By setting the list of crucial carbon emission units and the total amount of quotas, Other core policy objectives focus on promoting green development, promoting harmonious coexistence between man and nature, promoting the adjustment and upgrading of China's industrial structure, and strengthening the strategic adjustment of industrial structure.

Among the policy objectives of green finance, the highest score of feature vector centrality is "promoting green development," such as "Decision," "Made in China 2025," and "Guiding Opinions on Accelerating the Establishment and Perfection of Green and Low Carbon Circular Development Economic System" issued by the State Council in February 2021, indicating that the Chinese government has fully realized that innovation drive is the core driving force for realizing green finance, and proposed to vigorously develop green finance and green trading market mechanism, improve green standards, and ensure dual carbon The remaining core policy objectives are to promote high-quality economic development, promote sustainable development strategy, promote carbon emission reduction, accelerate the transformation of traditional industries, improve financing convenience and promote enterprises to assume social responsibility. These policy objectives reflect the main line of the Chinese government's efforts to promote green finance. The relevant policy objectives of promoting green finance

**Table 3. Promoting green financial policy objects (Chinese words) (Top 10).**

| | Green financial policy object | | | | | | Green financial policy behavior | | | | |
|---|---|---|---|---|---|---|---|---|---|---|---|
| Two-character word | Frequency/ times | Proportion/ % | Three-character word | Frequency/ times | Proportion/ % | Two-character word | Frequency/ times | Proportion/ % | Three-character word | Frequency/ times | Proportion/ % |
| Technology | 1526 | 1.45 | Model area | 1515 | 1.44 | Reform | 1674 | 1.73 | Informatization | 615 | 0.63 |
| Information | 1275 | 1.21 | Protected area | 1488 | 1.41 | Disclosure | 1648 | 1.70 | Digitization | 355 | 0.37 |
| Products | 1249 | 1.19 | New format | 1462 | 1.39 | Services | 1621 | 1.67 | Diversification | 252 | 0.26 |
| Industry | 1222 | 1.16 | Internet | 1435 | 1.36 | Development | 1594 | 1.64 | Digital intelligence | 224 | 0.23 |
| Science and technology | 1196 | 1.14 | Carbon account | 1382 | 1.31 | Innovation | 1568 | 1.62 | Carbon peaking | 166 | 0.17 |
| Market | 1143 | 1.09 | Experimental area | 1355 | 1.29 | Reform | 1054 | 1.09 | Modernization | 138 | 0.14 |
| Enterprise | 894 | 0.85 | Financial industry | 1102 | 1.05 | Supervision | 931 | 0.96 | Real-time | 119 | 0.12 |
| Resources | 869 | 0.83 | Big data | 741 | 0.70 | Perfect | 811 | 0.84 | Carbon neutralization | 100 | 0.10 |
| Government | 836 | 0.79 | New energy | 629 | 0.60 | Establish | 792 | 0.82 | Industrialization | 80 | 0.08 |
| Services | 762 | 0.72 | Agricultural products | 408 | 0.39 | Development | 773 | 0.80 | Initiative | 68 | 0.07 |

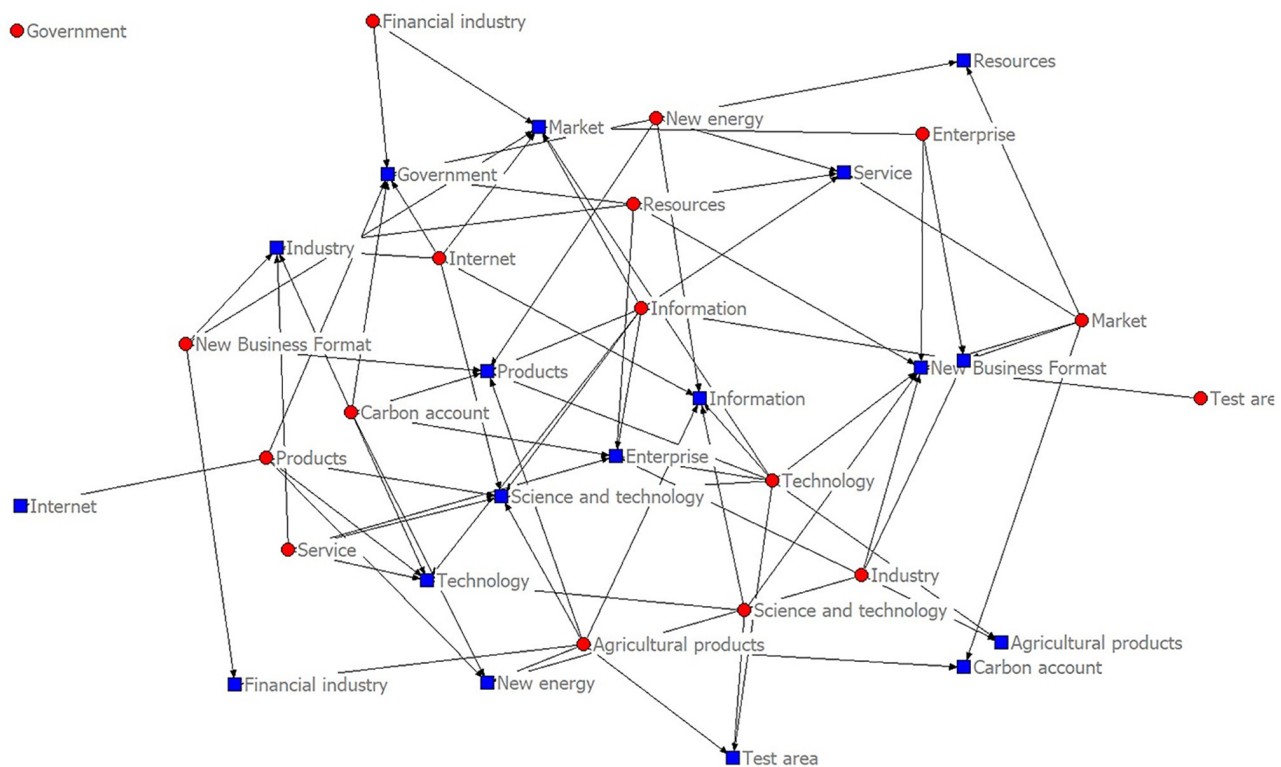

**Fig 6. Green financial policy content core words and network connection diagram.**

focus on promoting green innovation capability, promoting high-quality economic development, and promoting sustainable development strategy, among which are promoting carbon emission reduction, accelerating the transformation of traditional industries, improving financing convenience, and promoting enterprises to assume social responsibility are the key directions of green finance to promote green sustainable development of enterprises, thus helping to achieve the "double carbon" goal together with the carbon emission trading mechanism. The policy objectives of carbon emissions trading and green finance are shown in Table 4.

By comparison, we can find that carbon emission trading and green finance are consistent in policy objectives such as innovation-driven, industrial structure upgrading, and green development. The policy goal of the former focuses on the use of technology and resources to achieve green development by promoting the upgrading of industrial structure and the green transformation of enterprises; The latter's policy objectives are based on financial markets and enterprise resources, with innovation and industrial structure upgrading as the core driving force and green transformation as the main content. The distribution characteristics of the core policy objectives of the two policy systems respectively reflect the main control directions of enterprises, technologies, resources, and markets in the process of carbon emission reduction.

**2. Core policy tools.** Among the core policy tools for carbon emission trading, there are five command-controlled policy tools, namely, "collecting administrative fees for carbon trading," "controlling the market price of carbon emission trading," "establishing risk management mechanism and information disclosure system," "strengthening data management" and

**Table 4. Core policy objectives.**

| rank | Carbon emission trading | | Green finance | |
|---|---|---|---|---|
| | Name | E | Name | E |
| 1 | Promote green innovation of enterprises | 0.356 | Promote green development | 0.326 |
| 2 | Promote green development | 0.314 | Promote high-quality economic development | 0.266 |
| 3 | Promote carbon emission reduction | 0.305 | Promote the strategy of sustainable development | 0.264 |
| 4 | Promote the adjustment and upgrading of China's industrial structure | 0.287 | Promote carbon emission reduction | 0.256 |
| 5 | Strengthen the strategic adjustment of industrial structure | 0.281 | Accelerate the transformation of traditional industries | 0.223 |
| 6 | Forced the advanced transformation of the industrial structure | 0.277 | Improve financing convenience | 0.222 |
| 7 | Make up for the weak carbon market mechanism | 0.256 | Promote corporate social responsibility | 0.216 |
| 8 | Promote the carbon neutral technology innovation | 0.199 | Promote the transformation of the real economy | 0.213 |
| 9 | The Relationship between Stability and Controllability and Market Floating | 0.197 | Guide the allocation of financial resources | 0.195 |
| 10 | Handle the relationship between economic development and ecological protection | 0.189 | Promote green transformation | 0.192 |

Note: According to the centrality of feature vectors of core policy objectives, the top 10 are taken and sorted out. E denotes the centrality of the eigenvector.

"promoting and strengthening policy convergence"; There are three market incentive policy tools, namely, "carrying out carbon emission quota allocation and settlement," "improving CCER offset ratio quota" and "maintaining the healthy development of the market"; There are two public participation policy tools, namely, "establishing public participation procedures" and "encouraging the public and news media to supervise." This shows that the carbon emission trading mechanism mainly promotes carbon emission reduction under the supervision of public participation by improving the trading system, building a pilot market, and building a cooperation and trading platform. Among the core policy tools of green finance, there are four market incentive policy tools, namely, "carrying out standardized pilot demonstrations," "enriching financial products and service standards such as green credit and bonds," "exploring environmental rights trading methods" and "supporting green projects"; There are two public participation policy tools, namely, "Promoting Green Financial Reform and Innovation Pilot Zone" and "Perfecting and Refining the Existing Regulations on Public Participation in EIA"; There are four command-controlled policy tools, namely, "Promoting Green Financial Reform and Innovation Experimental Zone," "Coordinating with Green Fiscal Policy and Enriching Green Credit," "Financial Products and Service Standards such as Bonds" and "Strengthening the Development of Industry Standards". There are four market incentive policy tools, namely, "carrying out standardization pilot demonstration," "maintaining the healthy development of the market," "exploring the trading mode of environmental rights and interests," and "supporting the construction of green projects"; There are two public participation policy tools, namely, "establishing public participation procedures" and "perfecting and refining the existing regulations on public participation in EIA." This shows that China's green financial policy mainly promotes pilot areas, market construction, and public support, establishes and improves the implementation system of green finance and carbon emission trading policies, and shapes a

good market environment, thus providing endogenous power and various means to achieve the dual-carbon goal. The core policy tools of carbon emission trading and green finance are shown in Table 5.

By comparison, it is found that the use of core policy tools between carbon emission trading and green finance reflects that they can reduce greenhouse gas emissions, promote innovation, upgrade industrial structure, and contribute to realizing global emission reduction targets and sustainable development. Both of them not only have a high degree of consistency but also show their unique behavioral logic. The core behavioral logic of carbon emission trading policy tools lies in "pilot market" or "carbon market", which is embodied in the fact that in order to ensure the stability and effective operation of the carbon trading market, the government, enterprises and investors need to work together to strengthen supervision and cooperation in order to achieve sustainable economic, social and environmental development; The core behavioral logic of the use of green financial policy tools lies in "the comprehensive use of various tools", which is embodied in the fact that with the establishment of the green financial reform and innovation pilot zone and the standardized pilot demonstration operation framework, the domestic green credit, bonds, insurance, securities and other market operation mechanisms are becoming more and more perfect, and the attention of all localities to the expansion of carbon financial market and carbon market price is also increasing, which can encourage the government to further innovate the supervision mode, maintain the healthy

**Table 5. Core policy tools.**

| rank | Carbon emission trading | | | Green finance | | |
|---|---|---|---|---|---|---|
| | Name | E | Type | Name | E | Type |
| 1 | Carry out the allocation and settlement of carbon emission quotas | 0.268 | Market incentive type | Establish a public participation mechanism | 0.210 | Public participation type |
| 2 | Maintain the healthy development of the market | 0.251 | Market incentive type | Promote green financial reform and innovation pilot zone | 0.203 | Command control type |
| 3 | Carry out standardization pilot demonstration | 0.223 | Market incentive type | Carry out standardization pilot demonstration | 0.202 | Market incentive type |
| 4 | Improve CCER offset proportional quota | 0.216 | Command control type | Coordinate with green fiscal policy | 0.201 | Command control type |
| 5 | Establish public participation procedures | 0.190 | Public participation type | Maintain the healthy development of the market | 0.199 | Market incentive type |
| 6 | Levy administrative fees for carbon trading | 0.184 | Command control type | Enrich financial products and service standards such as green credit and bonds | 0.199 | Command control type |
| 7 | Encourage the public and news media to supervise | 0.174 | Public participation type | Explore the trading mode of environmental rights and interests | 0.195 | Market incentive type |
| 8 | Establish risk management mechanism and information disclosure system | 0.169 | Command control type | Improve and refine the existing regulations on public participation in EIA | 0.195 | Public participation type |
| 9 | Strengthen the data management | 0.159 | Command control type | Support the construction of green projects | 0.189 | Market incentive type |
| 10 | Promote and strengthen policy convergence | 0.155 | Command control type | Strengthen the development of industry standards | 0.189 | Command control type |

Note: According to the feature vector centrality of core policy tools, the top 10 are taken and sorted out. E denotes the centrality of the eigenvector.

operation of the market, and exert the emission reduction effect while improving the service efficiency of green financial market.

## Discussion

### Policy synergy analysis

We study the synergy between carbon emissions trading and the green financial policy system using quantitative analysis of differences, visualization of a policy network, and qualitative analysis of policy text.

There are structural asymmetries in carbon emissions trading and green financial policy systems, and the use of public participation policy tools in the two policy systems is relatively weak. The use of public participation policy tools for carbon emission trading only accounts for 11.82%, and the proportion of command control policy tools and market incentive policy tools is 43.64% and 44.55%, respectively; Only 11.91% of the public participation policy tools of green finance are used, and 43.40% of the command control policy tools and 44.68% of the market incentive policy tools are used respectively. The preferences for carbon emission trading and green financial policy tools are shown in Table 6.

We use the non-parametric test method to analyze the differences in using policy tools under different groups. Taking the policy system as the grouping variable, we test the number of policy tools of the two policy systems, and the results are shown in Table 7. The Z value of the Mann-Whitney test was -1.887, $p = 0.059 > 0.05$, and there was no significant difference. This means that the Chinese government has formed a relatively consistent behavior pattern in choosing policy tools to promote carbon emission trading policy and green finance. This behavior pattern does not have significant differences between the policy systems, but the difference in the number of policy tools reflects the primary and secondary relationship between the two.

Kruskal-Wallis test showed that H value was 0.834, $p = 0.659 > 0.05$. The results show no significant difference in the combination of policy tools and their quantitative composition between the two policy systems. However, there is a particular preference for choosing different types of policy tools. As shown in Table 8, in the distribution of different types of policy tools in the combined policy system, the quantitative relationship between command and

**Table 6. Preferences for the use of policy tools.**

| Type | Carbon emission trading | | Green finance | | After the merger | |
|---|---|---|---|---|---|---|
| | Frequency/ times | Frequency/% | Frequency/ times | Frequency/% | Frequency/ times | Frequency/% |
| Command control | 48 | 43.64 | 65 | 41.67 | 102 | 43.40 |
| Market incentive | 49 | 44.55 | 67 | 32.95 | 105 | 44.68 |
| Public participation | 13 | 11.82 | 24 | 15.38 | 28 | 11.91 |
| Total | 110 | 100 | 156 | 100 | 235 | 100 |

**Table 7. MannWhitney test results with policy system as grouping variables.**

| variable | Median of policy system M ($P_{25}$, $P_{75}$) | | MannWhitney test U value | Mann Whitney test Z value | P |
|---|---|---|---|---|---|
| | Carbon emissions trading (n = 3) | Green finance (n = 3) | | | |
| Frequency | 48.0 (49.0, 53.0) | 65.0 (67.0, 24.0) | 72.000 | -1.887 | 0.059 |

**Table 8. Kruskal-Wallis test results with policy types as grouping variables.**

| variable | Median of policy instruments M (P$_{25}$, P$_{75}$) | | | Kruskal-Walli: Test statistic H value | P |
|---|---|---|---|---|---|
| | Command Controll (n = 2) | Market incentive type (n = 2) | Public participation type (n = 2) | | |
| Frequency | 54.0 (48.0, 65.0) | 52.0 (49.0, 67.0) | 18.5 (13.0, 24.0) | 0.834 | 0.659 |

control policy tools, market incentive policy tools, and public participation policy tools is about 2.92: 2.8: 1. This quantitative relationship is 3.7: 3.8: 1 and 3.6: 3.8: 1 in the two independent policy systems, forming a differential pattern similar to a stepped distribution. The Kruskal-Wallis test was performed on three types of policy instruments with policy types as grouping variables, and the results are shown in Table 8.

As shown in Table 7, in the distribution of different types of policy tools in the consolidated policy system, the quantitative relationship among environmental policy tools, supply policy tools, and demand policy tools is about 2.9: 1.7: 1, which is 2.8: 1.3: 1 and 3: 2: 1 respectively in the two independent policy systems, forming a differential order pattern similar to stepped distribution.

## Qualitative analysis of policy texts

Through the descriptive analysis of text content, we can further understand the tendency to use policy tools behind complex semantics. We further analyze the synergistic characteristics of the two policy systems by using policy tools under the same policy objectives and different policy tools.

**1. Synergistic characteristics of policy tools under the same policy objectives.** Carbon emission trading and green finance have the same core policy objectives in promoting green innovation, upgrading industrial structure, and realizing green and low-carbon development. Table 9 lists the two policy systems' corresponding policy tools for the three objectives. The top five policy tools with feature vector centrality under each policy objective are selected for comparative analysis.

Table 10 lists the corresponding policy tools in the two policy systems under the same policy objectives. The top five policy tools with feature vector centrality under each policy objective are selected for comparative analysis.

From the results in Tables 8 and 9, it can be seen that the two policy systems have adopted "maintaining the healthy development of the market" and "establishing a coordinated working mechanism" as the top five policy tools, which shows that the government has realized that the market and system are the key elements to promote green innovation in its innovation policy, and attaches great importance to the top-level design and departmental coordination of policy implementation. Regulating the government involved in enterprises should strengthen supervision and supervision, ensure the fairness and transparency of the carbon trading market, and establish a reasonable carbon pricing mechanism. Secondly, enterprises should actively participate in the carbon trading market, reduce carbon emissions, and use market mechanisms to achieve economic benefits. At the same time, enterprises should also improve the transparency of environmental disclosure, provide investors with reliable ESG information, and stimulate the vitality of innovation subjects.

The two policy systems have adopted "carrying out standardization pilot demonstration" as one of their top five policy tools. There are apparent differences in the use of core policy tools. Carbon emission trading carries out carbon emission quota allocation and settlement, standardization pilot demonstration, and optimizes the primary environment for green financial

**Table 9. Comparison of policy instruments corresponding to policy objectives in the two policy systems.**

| Rank | Carbon emission trading | | | Green finance | | |
|---|---|---|---|---|---|---|
| | Name | E | Type | Name | E | Type |
| 1 | Carry out the allocation and settlement of carbon emission quotas | 0.268 | Market incentive type | Establish a public participation mechanism | 0.210 | Public participation type |
| 2 | Maintain the healthy development of the market | 0.251 | Market incentive type | Promote green financial reform and innovation pilot zone | 0.203 | Command control type |
| 3 | Carry out standardization pilot demonstration | 0.223 | Market incentive type | Carry out standardization pilot demonstration | 0.202 | Market incentive type |
| 4 | Improve CCER offset proportional quota | 0.216 | Command control type | Coordinate with green fiscal policy | 0.201 | Command control type |
| 5 | Establish public participation procedures | 0.190 | Public participation type | Maintain the healthy development of the market | 0.199 | Market incentive type |
| 6 | Levy administrative fees for carbon trading | 0.184 | Command control type | Enrich financial products and service standards such as green credit and bonds | 0.199 | Command control type |
| 7 | Encourage the public and news media to supervise | 0.174 | Public participation type | Explore the trading mode of environmental rights and interests | 0.195 | Market incentive type |
| 8 | Establish risk management mechanism and information disclosure system | 0.169 | Command control type | Improve the Regulations on Public Participation in EIA | 0.195 | Public participation type |
| 9 | Strengthen data management | 0.159 | Command control type | Support the construction of green projects | 0.189 | Market incentive type |
| 10 | Promote and strengthen policy convergence | 0.155 | Command control type | Strengthen the development of industry standards | 0.189 | Command control type |

development. Green finance establishes a public participation mechanism and promotes green and low-carbon consumption transformation. By strengthening market supervision or issuing administrative orders, the policy tools adopted by the two policy systems complement each other.

The two policy systems have adopted "leading green and low-carbon cities," "launching personal carbon emission reduction accounts," "encouraging the public and news media to supervise," and "establishing public participation procedures" in the top five policy tools. From the perspective of policy objectives, the two policy systems reflect the same goal of promoting carbon emission reduction by promoting green innovation, industrial structure upgrading, and realizing green and low-carbon development. From the perspective of the use of policy tools, the core policy behaviors of the two policy systems converge, among which carbon finance is an integral part of green finance. Various financial institutional arrangements and transactions to reduce greenhouse gas emissions include trading and investing carbon emission rights and their derivatives, investment and financing of low-carbon project development, and other related financial intermediary activities. Therefore, carbon finance is an essential combination of green finance and carbon emission trading and has become the two's core and consistent policy tool. However, the two have the characteristics of difference, complementarity, and consistency in the use of their unique policy tools, which reflects a certain extent that the mature carbon emission trading market and the green financial market should have the characteristics of the perfect legal basis, complete supporting system, high market participation, and good market order. However, green financial policy pays more attention to the effectiveness of

**Table 10. Comparison of the use of policy tools under the same policy objectives.**

| Promote green innovation | | Promote the upgrading of industrial structure | | Achieve green and low-carbon development | |
|---|---|---|---|---|---|
| Carbon emission trading mechanism | Green finance | Carbon emission trading mechanism | Green finance | Carbon emission trading mechanism | Green finance |
| Formulate guiding principles and policies | Promote green financial reform and innovation pilot zone | Carry out the allocation and settlement of carbon emission quotas | Support green environmental protection enterprises | Leading green and low-carbon cities | Establish a public participation mechanism |
| Maintain the healthy development of the market | Supervise high energy-consuming enterprises | Carry out standardization pilot demonstration | Supervise high energy-consuming enterprises | Launch a personal carbon emission reduction account | Promote the transformation of green and low-carbon consumption |
| Carry out the allocation and settlement of carbon emission quotas | Support the construction of green projects | Carry out carbon budget management of key emission units | Carry out standardization pilot demonstration | Encourage the public and news media to supervise | Coordinate with green fiscal policy |
| Levy administrative fees for carbon trading | Maintain the healthy development of the market | Formulate guiding principles and policies | Strengthen the development of industry standards | Establish public participation procedures | Improve the Regulations on Public Participation in EIA |
| Establish a coordination mechanism | Establish a coordination mechanism | Levy administrative fees for carbon trading | Optimize the basic environment for green financial development | Strengthen the connection between carbon market and voluntary emission reduction mechanism | Launch a personal carbon emission reduction account |

policies, and market participants pay more attention to the policy logic of reasonable carbon price level and market effectiveness.

On the one hand, carbon trading depends on the support of the green financial market, which is not only reflected in the leading role of carbon finance business in promoting green development and carbon emission reduction of enterprises but also reflected in the guiding role of the construction and improvement of the green financial market in green innovation and industrial upgrading. On the other hand, the ultimate goal of promoting the development of a carbon emission trading market is to supplement the green financial business by entirely using the carbon trading market, carbon emission rights, and low-carbon technology. For example, the inclusion of high-emission enterprises in the carbon emission trading market is conducive to forcing high-emission enterprises to achieve carbon emission reduction targets by learning, drawing lessons from, and absorbing advanced production and green management experience, actively fulfilling green standards and participating in green financial business, and actively fulfilling the contract can also force enterprises to achieve green transformation. Based on this, we interpret the policy path of carbon emission reduction as "supporting the green financial business with carbon trading market tools and promoting the perfection and supervision of carbon trading market with the green financial market.

**2. Synergistic characteristics of different types of policy tools.** Order-controlled market incentive public participation, improve carbon trading system, collect administrative fees,

strengthen performance supervision, carry out pilot demonstrations, encourage market transactions, develop market tools, encourage public supervision, information and data management, encourage personal participation, coordinate with green finance, enrich products and services, strengthen industry standards, carry out pilot demonstrations, maintain market development, explore trading methods, promote public participation, promote green projects, promote carbon inclusiveness, order-controlled market incentive public participation, maintain healthy market development, stimulate the vitality of innovative subjects, attach importance to top-level design, establish coordination mechanism, lead green and low-carbon development, and actively fulfill green standards, order-controlled market incentive public participation Good carbon trading system collects administrative fees, strengthens performance supervision, conducts pilot demonstrations, encourages market transactions to develop market tools, encourages public supervision, information and data management, encourages personal participation and green finance to coordinate, enriches products and services, strengthens industry standards, conducts pilot demonstrations, maintains market development, explores trading methods, promotes public participation, promotes green projects, promotes carbon inclusiveness, command-controlled market incentives, public participation, maintains healthy market development, stimulates the vitality of innovative subjects, attaches importance to top-level design, establishes coordination mechanisms, leads green and low-carbon development, and actively implements green standards. There are three typical policy tool themes: system, taxation, and supervision; market incentive, including pilot, transaction, and tool; and public participation, including supervision, management, and encouragement. Based on building a collaborative framework of carbon emission reduction policies, as shown in Fig 7, the "supporting" role of the carbon emission trading mechanism and the "normative" role of green finance are further analyzed.

As shown in Fig 7, from the perspective of command-control policy tools, the rational allocation of carbon emission quotas can be realized by establishing a carbon emission trading market and improving its operation and supervision system. High-energy-consuming

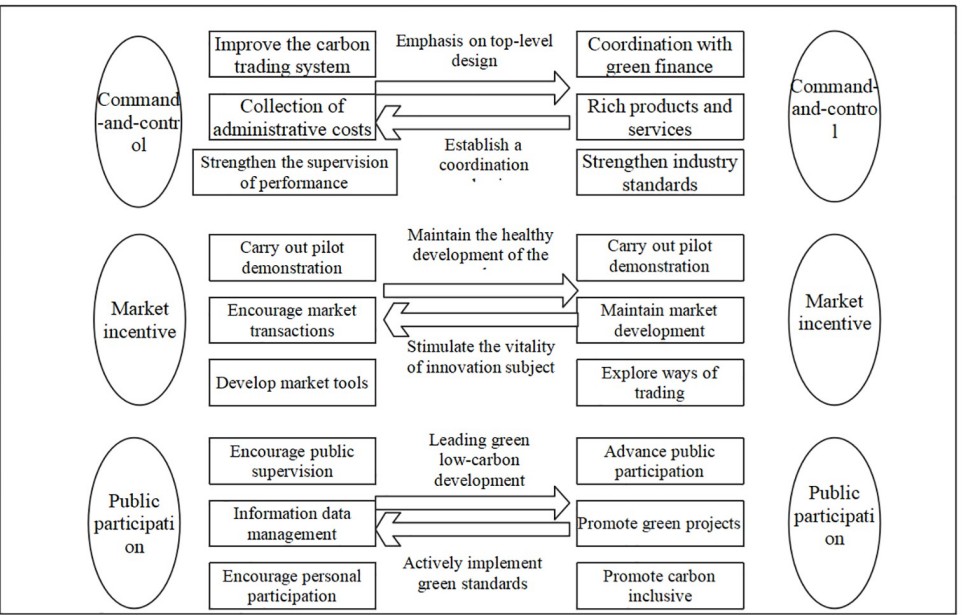

**Fig 7. Collaborative emission reduction framework of green finance and carbon emission trading policy.**

enterprises can be forced to purchase quotas from low-energy-consuming enterprises, thus promoting critical emission units' mandatory emission reduction performance. The green financial market aims to build a green financial order. For example, the Property Insurance Supervision Department of the State Administration of Financial Supervision issued the Guiding Opinions on Promoting the High-quality Development of Green Insurance (Draft for Soliciting Opinions) to insurance companies and insurance associations, which put forward requirements for the critical tasks on the debt side, the essential tasks on the investment side, the ability support and the job guarantee of the development of green insurance. Guide the circulation of market funds to low-energy-consuming enterprises through various green financial instruments. In 2021, the first performance period of the national carbon market started and opened for trading, and the supervision and management of carbon emission trading have been further strengthened. Many subjects, such as regulatory agencies, critical emission units, technical service agencies, national carbon emission registration agencies, trading institutions, settlement banks, and institutions and individuals that meet the relevant national trading rules, all participate in the market. Therefore, a scientific and reasonable regulatory system is needed to prevent and control the risk of "washing green" and defaulting.

From the perspective of market incentive policy tools, both policy systems emphasize improving the pilot market and optimizing emission reduction targets by increasing market transactions and market competition. Among them, the carbon emission trading policy is based on exerting the "local market effect." By participating in the pilot market and the national carbon emission trading market, it can achieve the quota target and enhance the momentum of emission reduction. In July 2021, China's carbon emission trading market officially started online trading. As of the end of the transaction on November 24, 2022, the cumulative turnover of carbon emission quotas exceeded 200 million tons, and more than half of the critical emission units participated in the transaction. This shows that with the gradual advancement of the construction of the national carbon emission trading market, both carbon quotas and certified voluntary emission reductions (CCER) can increase the liquidity and activity of the carbon market from the perspective of market trading and improve the emission reduction capacity of related enterprises. The policy tools used in green financial policy include green credit, green bonds, carbon trading, ESG funds, green insurance, green gold branch, green trust, and so on. The multi-level green financial system covers a variety of products. With the promotion of the "double carbon" goal, various green financial assets have begun to take shape, guiding enterprises to achieve carbon emission reduction through the allocation of green financial assets and achieve specific emission reduction targets at a lower cost.

## Challenges in the collaborative process

From the public participation policy tools perspective, the two policy systems actively attract more people to participate in activities such as carbon neutralization and emission reduction. The policy tools used include implementing a personal investment system, establishing a public access system, promoting green projects, replicating and popularizing the experience of first-in-first-trial, and encouraging individuals to participate in green finance and carbon trading. Under the carbon emission trading policy, there is a precedent for individuals to enter the market. As an innovative carbon finance model, carbon inclusive will transform individuals' low-carbon behavior into economic value and activate market liquidity. As an energy-saving and emission-reduction behavior, carbon inclusiveness increases the actual demand for carbon sink projects. Green finance policy is issued to individual investors, broadening the channels for the public to participate in green investment and expanding green finance's social effects. Both can strengthen the public's awareness of green responsibility investment to protect the

environment and support green development. Analyzing the synergy of policy tools under different systems shows that carbon emission trading policies are characterized by supervision, adjustment, and platform. In contrast, green financial policy tools are characterized by guidance, public welfare, and externality, constituting a "dual synergy" carbon emission reduction policy that is complementary, embedded, and integrated. Based on the perspective of policy tool goal, we find the following problems and challenges in the process of policy synergy:

First, the choice of policy tools is diversified, but there is a significant structural imbalance. The low degree of adaptation between tools and goals restricts the effectiveness of policies, which will lead to a certain degree of deviation between realistic goals and policy expectations, and cannot meet the policy needs of the current market-driven stage. Green financial policy has a solid positive external effect in supporting enterprises' green development. To a certain extent, green financial products have the characteristics of public goods, and their essence is to protect and improve the ecological environment. Green financial products and services will incur costs, while consumers will not incur any costs when enjoying the benefits of ecological environment improvement. Therefore, there may be a need for more supply. This is consistent with Volz et al.'s (2015) and Soundarrajan & Vivek (2016) research [50, 51]. There are structural imbalances such as the overuse of command tools, the relative underuse of symbol and exhortation tools, and the serious absence of capacity-building tools, incentive tools, and system change tools. We can further define the carbon intensity based on the relevant theories of new institutional economics, stipulate the total amount of carbon intensity in the assets of all financial institutions, and issue licenses for financial institutions. By establishing a perfect standard system, information disclosure, and other mechanisms, we can solve the dilemma of green financial policy through marketization.

Second, the government's attention configuration is more concentrated. Green finance has a 'free rider 'behavior. From the perspective of government decision-making, government decision-makers take green innovation and green development as policy objectives and draw a blueprint for the promotion of dual-carbon goals and work, providing direction for action. However, it should be noted that green projects usually have low returns and long payback periods and need additional certification for green projects. The financial "profit-seeking" feature will further strengthen financial institutions' "free rider" behavior in developing green finance. We should pay attention to the matching of policy objectives and tools, so as to 'improve quality and efficiency' while taking into account the comprehensive governance process of supervising high-energy-consuming enterprises, supporting green project construction, maintaining the healthy development of the market, and establishing a coordinated working mechanism. This further extends the research content of Guo & Tan (2023) and Dunlap & Laratte (2022) [52, 53]. Therefore, we should prioritize making up for market failure, strengthening policy support for technological innovation and industrial upgrading by coordinating carbon trading and green financial policies, and improving "free rider behavior."

Third, enhance policy orientation. The characteristics of policy tool selection between different objectives are similar, which can easily lead to policy failure. Some policy tools with a high frequency of application are not clear enough, resulting in uncertainty in the implementation effect. For example, market-based policy tools are mostly supported, strengthened, and encouraged in expression, and there are no clear implementation provisions such as support mechanisms, forms, and channels, resulting in uncertainty in the degree and effect of implementation. Carbon finance is an essential product that reflects the synergy between green finance and carbon emission trading. It is still in the stage of product development, and there are risks such as high cost and abnormal fluctuation of carbon price. Some problems in carbon finance, such as complex technology, long cycle, low profit, high policy risk, asymmetric information, lead to insufficient motivation for carbon finance to a certain extent. Green finance is

still dominated by traditional project financing and green credit, and its product structure and service mode are single. This complements the research contents of Zhou & Li (2019) and Simon et al. (2012) [15, 54]. We can further innovate green financial products and services, implement environmental marketization means, improve the pilot design, operation, and performance of carbon emissions trading system, and seek innovative breakthroughs in project construction and management.

## Research conclusions and management enlightenment

### Research conclusions

Based on the matching network of "policy objective-policy tool," the policy texts on promoting carbon emission trading and green finance are decomposed in refined grains, and the primary objective orientation, core policy path, policy network characteristics, and policy coordination of the two policy systems are systematically analyzed. It is found that carbon emission trading and green finance constitute a domestic and international "dual synergy" policy system through consistent policy objectives and complementary policy tools, and they converge in core policy objectives such as innovation-driven, capacity adjustment, and green development. The export-oriented, pioneering, and platform-oriented characteristics of carbon emission trading policy complement the inward-oriented, adjusted, and regulatory characteristics of green financial policy tools. The difference in the number of policy tools used by the two reflects the primary and secondary characteristics of "green finance is the mainstay, supplemented by carbon emission trading," which reflects the evolution trend of China's carbon emission reduction policy to "green financial policy is the mainstay, carbon trading, and green financial policy interact synergistically." The proposal for carbon emission reduction is related to the previous policy inertia.

### Management enlightenment

Based on the synergistic characteristics of carbon emission reduction policies, we put forward the following policy suggestions. Based on the quantitative analysis of policy content, we believe that China should optimize the carbon emission trading mechanism and green financial policy tools from the following three aspects, and play a synergistic effect to achieve the dual carbon goal:

First, moderately increase the proportion of command-and-control policy tools. Based on the existing policy system, the policy objectives and policy tools are systematically optimized according to the interactive characteristics of carbon emission reduction: First, focus on the typical emission reduction policy theme of green finance and carbon emission trading, avoid the conflict of policy objectives, and strengthen the coordination of policy tools. Focus on the standard policy objectives of promoting green innovation, upgrading industrial structure and realizing green and low-carbon development, improving green financial instruments while promoting the construction of a carbon emission trading market, and paying attention to improving the participation, activity, and flexibility of carbon emission trading market while promoting the implementation scope of green finance, creating favorable conditions for the realization of domestic carbon emission reduction targets. Second, promote the construction of a unified national carbon emission trading market, realize the cross-regional market operation mechanism, and further expand the scope of the green financial reform and innovation pilot zone to promote the coordinated play of emission reduction between regional and interregional carbon emission trading and green financial policies, achieve target integration and form standard norms, and strengthen the timeliness and consistency of emission reduction policies. For market incentive environmental policy tools, we should continue to carry out

technology promotion, government procurement, pilot/demonstration projects, etc., and accelerate the applicability promotion of zero-carbon building technology. For example, in the field of transportation, the use of market incentive policy tools should be improved, such as stimulating the market demand for new energy vehicles and helping the market-driven transformation of the new energy vehicle industry.

Second, improve the policy system of target coordination. Maintain the cohesion of different types of environmental policies. For example, the industrial sector should comprehensively use the financial subsidies, science and technology investment, and incentive mechanisms of the sub-categories of market-based incentive policy tools, the industrial layout, structural adjustment, tax incentives, financial support, and access rules of the sub-categories of command-and-control policy tools, and the incentives of public participation-based policies to accelerate the improvement of policy tools in different fields. We should actively promote the application of clean technology, formulate stricter control rules and access rules to promote energy efficiency, and formulate multi-type environmental policies of price, tax, and financial support. The construction of a target synergy policy system helps solve the problem of insufficient coordination of policy objectives and promotes communication and commonness between the two policy systems. For example, in emission reduction, green financial policies should solve the problems of complex and expensive financing for green projects and strengthen the forecasting preferences of enterprises and consumers. The carbon emission trading policy aims to realize the allocation and performance of carbon emission quotas. There is target synergy in the emission reduction process, so we should further ensure the smooth implementation of policies from scientific, reasonable, and feasible aspects, that is, promote the enthusiasm of enterprises to reduce emissions by improving green financing means and carbon emission trading management. In addition, the duplication between policies should be fully considered, and the effectiveness of carbon emission reduction should be improved through the synergy between targets.

Third, promote the optimization and balance of policy tools. Improve the accuracy and enforcement of various types of policy tools to enhance the breadth and threshold standards for the application of access rule policy tools. Formulate clear support policies and preferential tax policies, stimulate market demand, and effectively play a role in promoting green development. With the in-depth implementation of the dual-carbon goal, significant changes have taken place in the development of different industries, and the use structure of carbon emission trading and green financial policy tools should be further optimized. Improve the application ratio of public participation tools, improve the awareness of carbon emission reduction of the whole people by improving the enthusiasm of public participation, and further enhance the emission reduction effect. We can optimize the internal structure of command control and tools by improving the public participation system and standards to reduce public participation barriers and enhance the consistency of carbon emission reduction targets. The government should combine quota management with market transactions, strengthen the guiding role of the market, and continue to promote green innovation and industrial upgrading at the technical level to provide a basis for enterprises to implement emission reduction targets and green development tasks and improve China's overall carbon emission reduction capacity. By perfecting the market system and social participation and strengthening the development of industry standards, we can make up for the problem of insufficient use of public participation tools.

## Limitations and prospects

Based on the three-dimensional analysis framework of "policy subject-policy tool-policy theme," we analyze the multi-dimensional structure of information confidentiality policy from

the form and content characteristics, which has specific practical significance for studying the development status of carbon emission reduction targets in China. However, limited by the research perspective, policy samples, and research content, this paper still has limitations in many aspects. In the future, it is necessary to expand the analysis dimension, integrate policy objectives, policy objects, and other elements, and focus on cross-analysis among elements. At the same time, we can further collect local and foreign policy data sets and deeply compare the structural characteristics of green finance and carbon emission trading policies between central and local, domestic and foreign. Conduct inter-governmental analysis. Introduce a policy evaluation model, determine multiple indicators, and analyze the emission reduction effectiveness of green financial and carbon emission trading policies from multiple stages of policy formulation and implementation.

## Supporting information

**S1 Data.**
(ZIP)

## Acknowledgments

We thank the Zhejiang Sci-Tech University for foundation. We also want to thank the editors and the reviewers for their instructions and comments.

## Author Contributions

**Conceptualization:** Hongjian Yu.

**Data curation:** Hongjian Yu.

**Formal analysis:** Hongjian Yu.

**Funding acquisition:** Xiufan Zhang.

**Methodology:** Hongjian Yu.

**Resources:** Hongjian Yu, Xiufan Zhang.

**Writing – original draft:** Hongjian Yu.

**Writing – review & editing:** Xiufan Zhang.

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
