## [Decision Letter · Decision Letter 0]

8 Jan 2024

PONE-D-23-43341Are carbon emissions trading and green financial instruments synergistic? -Comprehensive quantitative research based on content analysisPLOS ONE

Dear Dr. Zhang,

Thank you for submitting your manuscript to PLOS ONE. After careful consideration, we feel that it has merit but does not fully meet PLOS ONE’s publication criteria as it currently stands. Therefore, we invite you to submit a revised version of the manuscript that addresses the points raised during the review process.

We look forward to receiving your revised manuscript.

Kind regards,

Pengyu Chen

Academic Editor

PLOS ONE

Journal Requirements:

Reviewers' comments:

Reviewer's Responses to Questions

**Comments to the Author**

1. Is the manuscript technically sound, and do the data support the conclusions?

Reviewer #1: Yes

Reviewer #2: Yes

2. Has the statistical analysis been performed appropriately and rigorously? 

Reviewer #1: Yes

Reviewer #2: Yes

3. Have the authors made all data underlying the findings in their manuscript fully available?

Reviewer #1: Yes

Reviewer #2: Yes

4. Is the manuscript presented in an intelligible fashion and written in standard English?

Reviewer #1: Yes

Reviewer #2: Yes

5. Review Comments to the Author

Reviewer #1: This paper makes word frequency statistics according to the policy system, and summarizes the core objectives and tools of green financial policy and carbon emission trading policy. For the classification of policy tools, according to the characteristics of green finance policy and carbon emission reduction policy, this paper divides policy tools into ' command control type ', ' market incentive type ' and ' public participation type '.Based on the existing policy network, the carbon emission reduction policy system is constructed, and the future direction of using green finance and carbon emission trading policy to guide the development of coordinated emission reduction is analyzed. The article is innovative. In order to improve the quality of the article, we put forward the following suggestions :

1.The theoretical analyses in abstract of the article is overloaded, and it is recommended that some of them be reorganised or included in the introduction.

2.Literature review. The analysis of literature review is not systematic enough. It is suggested to add the latest research results and point out the gaps in the existing research, so as to show the innovation and necessity of this paper.

3.Further beautiful charts to make it more standardized. In particular, Figure 2 and Figure 3.There are many tables in this paper, and it is recommended to set the table as a three-line table and improve the format of the table.

4. Research methods and data processing. I suggest adding a key time node table for some policies. Analyze policies at different levels to highlight intergovernmental relations.

5. Results and Analysis. I propose to add a discussion section, move some of the analysis in the results to the discussion, and compare the results of the article with the existing literature to better illustrate the innovation and contribution of this article..

Reviewer #2: This paper studies the optimization of China's carbon emission trading and green finance policies. The applied data are reliable, the research methods are feasible, the research conclusions are enlightening, and three optimization strategies are obtained. However, the optimization strategy is too macro, and it is hoped that the author can make more elaboration on this aspect of "improving the content of the policy system, improving the coordination goal of the policy system, and promoting the optimization and balance of policy tools", so as to improve the pertinence and operability of the countermeasures. At the same time, the format and language of the paper need to be refined and modified to achieve the integration of academic and native English.

6. PLOS authors have the option to publish the peer review history of their article (what does this mean?). If published, this will include your full peer review and any attached files.

Reviewer #1: No

Reviewer #2: **Yes: **liu bangfan

---

## [Author Response · Author response to Decision Letter 0]

18 Jan 2024

Response to Reviewers

Thank you very much for the revision suggestions of the study, which have allowed us to gain a lot, and deepened more thinking to better improve our study. We hope that our revisions can make the quality of the article better. If there are any inappropriate revisions, we will continue to work hard. Hope to have the opportunity to be published. Thanks again! The following is the specific modification part and we mark the changes in red:

Reviewer #1: This paper makes word frequency statistics according to the policy system, and summarizes the core objectives and tools of green financial policy and carbon emission trading policy. For the classification of policy tools, according to the characteristics of green finance policy and carbon emission reduction policy, this paper divides policy tools into ' command control type ', ' market incentive type ' and ' public participation type '.Based on the existing policy network, the carbon emission reduction policy system is constructed, and the future direction of using green finance and carbon emission trading policy to guide the development of coordinated emission reduction is analyzed. The article is innovative. In order to improve the quality of the article, we put forward the following suggestions.

1.The theoretical analyses in abstract of the article is overloaded, and it is recommended that some of them be reorganised or included in the introduction.

Thanks for the valuable advice from the experts. We organize the abstract part of the article and explain the research purpose, methods, results and main contributions of the article. At the same time, part of the content is placed in the introduction. 

2.Literature review. The analysis of literature review is not systematic enough. It is suggested to add the latest research results and point out the gaps in the existing research, so as to show the innovation and necessity of this paper. 

Thanks for the valuable advice from the experts. We add the latest research results to the literature review, so as to summarize the existing research results and findings, and find out the shortcomings of the existing research, which better reflects the innovation of this research. 

3.Further beautiful charts to make it more standardized. In particular, Figure 2 and Figure 3.There are many tables in this paper, and it is recommended to set the table as a three-line table and improve the format of the table.

Thanks for the valuable advice from the experts. We have revised and embellished the charts of the full text, and modified the format of the full text according to the style and requirements of the PLos One journal. See the text for specific modifications.

4.Research methods and data processing. I suggest adding a key time node table for some policies. Analyze policies at different levels to highlight intergovernmental relations.

Thanks for the valuable advice from the experts. We increase the time node, policy keywords and network analysis of the policy to better quantify the policy text. 

5.Results and Analysis. I propose to add a discussion section, move some of the analysis in the results to the discussion, and compare the results of the article with the existing literature to better illustrate the innovation and contribution of this article.

Thanks for the valuable advice from the experts. We added the discussion section.

Reviewer #2:

This paper studies the optimization of China's carbon emission trading and green finance policies. The applied data are reliable, the research methods are feasible, the research conclusions are enlightening, and three optimization strategies are obtained. However, the optimization strategy is too macro, and it is hoped that the author can make more elaboration on this aspect of "improving the content of the policy system, improving the coordination goal of the policy system, and promoting the optimization and balance of policy tools", so as to improve the pertinence and operability of the countermeasures. At the same time, the format and language of the paper need to be refined and modified to achieve the integration of academic and native English.

Thanks for the valuable advice from the experts. We will make more elaborations on ' improving the content of the policy system, improving the coordination objectives of the policy system, and promoting the optimal balance of policy tools ', so as to improve the pertinence and operability of the countermeasures. 

At the same time, we refine and modify the format and language of the paper, and ask native English speakers to polish the language to achieve the integration of academic English and native English.

---

## [Decision Letter · Decision Letter 1]

29 Jan 2024

Are carbon emissions trading and green financial instruments synergistic? -Comprehensive quantitative research based on content analysis

PONE-D-23-43341R1

Dear Dr. Zhang,

We’re pleased to inform you that your manuscript has been judged scientifically suitable for publication and will be formally accepted for publication once it meets all outstanding technical requirements.

Kind regards,

Pengyu Chen

Academic Editor

PLOS ONE

Additional Editor Comments (optional):

Reviewers' comments:

Reviewer's Responses to Questions

**Comments to the Author**

1. If the authors have adequately addressed your comments raised in a previous round of review and you feel that this manuscript is now acceptable for publication, you may indicate that here to bypass the “Comments to the Author” section, enter your conflict of interest statement in the “Confidential to Editor” section, and submit your "Accept" recommendation.

Reviewer #1: (No Response)

Reviewer #2: All comments have been addressed

2. Is the manuscript technically sound, and do the data support the conclusions?

Reviewer #1: (No Response)

Reviewer #2: Yes

3. Has the statistical analysis been performed appropriately and rigorously? 

Reviewer #1: (No Response)

Reviewer #2: Yes

4. Have the authors made all data underlying the findings in their manuscript fully available?

Reviewer #1: (No Response)

Reviewer #2: Yes

5. Is the manuscript presented in an intelligible fashion and written in standard English?

Reviewer #1: (No Response)

Reviewer #2: Yes

6. Review Comments to the Author

Reviewer #1: Paper is ok.The manuscript has significantly improved as compared to the previous version. Indeed, the authors tried to improve it, and the main weaknesses are solved.

Thus, in my opinion, the manuscript is recommendable for publication..

Reviewer #2: The author of this article has made the revision according to the appraisal expert opinion, has reached the publication level, proposed the publication.

7. PLOS authors have the option to publish the peer review history of their article (what does this mean?). If published, this will include your full peer review and any attached files.

Reviewer #1: No

Reviewer #2: **Yes: **liu bangfan

---

## [Editor Report · Acceptance letter]

28 Feb 2024

PONE-D-23-43341R1 

PLOS ONE

Dear Dr. Zhang, 

I'm pleased to inform you that your manuscript has been deemed suitable for publication in PLOS ONE. Congratulations! Your manuscript is now being handed over to our production team.

Kind regards, 

on behalf of

Dr. Pengyu Chen 

Academic Editor

PLOS ONE